# Proteomic Signatures of the Serological Response to Influenza Vaccination in a Large Human Cohort Study

**DOI:** 10.3390/v14112479

**Published:** 2022-11-09

**Authors:** Shaohuan Wu, Smruti Pushalkar, Shuvadeep Maity, Matthew Pressler, Justin Rendleman, Burcu Vitrinel, Michael Carlock, Ted Ross, Hyungwon Choi, Christine Vogel

**Affiliations:** 1Center for Genomics and Systems Biology, New York University, New York, NY 10003, USA; 2Birla Institute of Technology and Science (BITS)-Pilani (Hyderabad Campus), Hyderabad 500078, India; 3Department of Infectious Diseases, College of Veterinary Medicine, University of Georgia, Athens, GA 30602, USA; 4Center for Vaccines and Immunology, University of Georgia, Athens, GA 30605, USA; 5Department of Medicine, Yong Loo Lin School of Medicine, National University of Singapore, Singapore 117599, Singapore

**Keywords:** influenza vaccination, serological response, serum proteomics, cohort analyses, seroconversion

## Abstract

The serological response to the influenza virus vaccine is highly heterogeneous for reasons that are not entirely clear. While the impact of demographic factors such as age, body mass index (BMI), sex, prior vaccination and titer levels are known to impact seroconversion, they only explain a fraction of the response. To identify signatures of the vaccine response, we analyzed 273 protein levels from 138 serum samples of influenza vaccine recipients (2019–2020 season). We found that levels of proteins functioning in cholesterol transport were positively associated with seroconversion, likely linking to the known impact of BMI. When adjusting seroconversion for the demographic factors, we identified additional, unexpected signatures: proteins regulating actin cytoskeleton dynamics were significantly elevated in participants with high adjusted seroconversion. Viral strain specific analysis showed that this trend was largely driven by the H3N2 strain. Further, we identified complex associations between adjusted seroconversion and other factors: levels of proteins of the complement system associated positively with adjusted seroconversion in younger participants, while they were associated negatively in the older population. We observed the opposite trends for proteins of high density lipoprotein remodeling, transcription, and hemostasis. In sum, careful integrative modeling can extract new signatures of seroconversion from highly variable data that suggest links between the humoral response as well as immune cell communication and migration.

## 1. Introduction

Influenza poses an ongoing challenge to public health, causing tens of millions of infections and tens of thousands of deaths annually in the United States (https://www.cdc.gov/flu/about/index.html, accessed on 20 September 2022). Due to its high mutation rate, it requires annual vaccination, typically consisting of inactivated components of representatives of the four main strains, A/H1N1, A/H3N2, B/Yamagata, and B/Victoria. However, the influenza vaccine is only ~50% effective on average (https://www.cdc.gov/flu, accessed on 4 November 2022), implying that about half of the recipients do not seroconvert, i.e., increase their antibody titer levels.

While plausible reasons for this inefficient vaccine response are known or suspected, identification of markers of vaccine efficacy are still an area of active investigation. For example, prior work identified genetic [1] or epigenetic factors [2] as well as pre-existing immunity that raises baseline antibody titers [3] caused by prior infection or vaccination [4]. In addition, age [5,6], obesity [7], and sex [8] are thought to modulate the response. Our prior work quantified the impact of age, body mass index (BMI), sex, race, prior vaccination status, baseline antibody titers, month of vaccination, vaccine dose, and the presence of comorbidities across a large cohort study and found that these factors combined can predict ~80% of the response to the vaccine, as measured by seroconversion [9]. Further, we used this model to compute a new, adjusted seroconversion value which accounts for the impact of these factors.

To identify new factors that impact vaccine efficacy, several studies examined blood samples from vaccine recipients. Goronzy et al. showed that the frequency of CD8(+) CD28(−) T cells is associated with the antibody response to the influenza vaccine in older individuals [10]. Duin et al. discovered that Toll-like receptor (TLR) induced expression of the B7 costimulatory molecules CD80 and CD86 in peripheral blood mononuclear cells (PBMCs) is associated with the antibody response to the influenza vaccine [11]. Later Panda et al. showed that decreased TLR levels in dendritic cells of older individuals is associated with the dysregulation of cytokines and a poor antibody response [12].

However, large-scale studies to identify molecular signatures of the immune response are still rare [13]. One study examined the transcriptome in human PBMCs and showed that the expression of nine genes is associated with seroconversion against the influenza vaccine after adjusting for the baseline antibody titers [14]. A more recent study using lectin microarrays and glycoproteomics found that the serum glycan Le^a^ is associated with the serological response to the influenza vaccine, and further, that Le^a^-binding proteins are enriched in the complement system [15].

To expand on these studies, we examined the abundance profiles of serum proteins from recipients of the split-inactivated influenza vaccine using mass spectrometry based proteomics in a large cohort study. To estimate the vaccine response, we used a measure of seroconversion as defined previously [16], consisting of the hemagglutination inhibition (HAI) measurements against the four vaccine strains from before and after vaccination (day 0 and day 28). We also matched protein levels with an adjusted seroconversion value that was corrected for the known impact of age, BMI, sex, prior vaccination status, month of vaccination, vaccine dose, baseline titer levels, and comorbidities [9]. We analyzed the data using different statistical methods and report robust results that suggest new factors modulating the vaccine response.

## 2. Methods

### 2.1. Sample Collection

The serum samples were procured from a large human cohort study conducted by Ted Ross at University of Georgia, Athens (IRB #3773). Specifically, we analyzed samples from 160 participants from the 2019–2020 influenza season. Participants had received the split-inactivated influenza vaccine. Information for all samples included participant age, body mass index (BMI), sex, race, comorbidities, prior vaccination status, month of vaccination, and vaccine dose. As per study design, participants chose the timing of the vaccination; however, older participants had been encouraged to receive the vaccine early in the season. Information for each sample also included results from hemagglutination inhibition (HAI) assays against each vaccine strain for both the baseline (day 0) prior to vaccination and day 28 post vaccination.

### 2.2. Estimates of Seroconversion

To estimate seroconversion, we used the procedure published in [16]. S train-specific seroconversion was calculated as the log_2_ ratio of the HAI titer levels at Day 28 versus Day 0. Composite (overall) seroconversion was calculated as the sum of the strain-specific seroconversion values for the four vaccine strains. For each sample, we obtained information for ‘adjusted seroconversion’ following the published procedure [9]. Adjusted seroconversion corrects for the impact of participant age, BMI, sex, race, comorbidities, prior vaccination status, baseline titer levels, month of vaccination, and vaccine dose. Therefore, adjusted seroconversion represents the response to the influenza vaccine while removing confounding factors. For example, as reported [9], Month of vaccination positively affected the response to the vaccine, likely due to concurrent influenza infections. This impact, as well as that of all other factors, was removed in the adjusted seroconversion [9].

### 2.3. Sample Preparation

We processed 225 serum samples including individual and quality control 160 samples as per the protocol described elsewhere (Allgoewer K, 2021). Briefly, 1 μL of serum sample (~70–80 μg protein) was lysed with 0.1% Rapigest (Waters, Milford, MA, USA) in 100 mM ammonium bicarbonate (Sigma, St. Louis, MO, USA) followed by denaturation at 95 °C for 5 min. Later, the samples were reduced using 5 mM dithiothreitol (DTT, Sigma) at 60 °C for 30 min, and alkylated with 15 mM iodoacetamide (Sigma) in the dark for 30 min at room temperature. The samples were subsequently quenched with 10 mM DTT with overnight digestion at 37 °C using Trypsin gold (Promega, Madison, WI, USA). The digestion was terminated and the surfactant was cleaved by treating the samples with 200 mM HCl (Sigma) for 30 min at 37 °C. The desalting of the samples was performed using Hypersep C-18 spin tips (Thermo Fisher Scientific, Waltham, MA, USA). The eluted peptides were dried under vacuum at room temperature using a Vacufuge Plus (Eppendorf, Enfield, CT, USA) and were resuspended in 5% acetonitrile with 0.1% formic acid (Thermo Scientific). The resulting peptides were quantified by fluorometric peptide assay kit (Thermo Fisher Scientific) before mass spectrometry analysis.

The samples were subjected to an EASY-nLC 1200 (Thermo Fisher Scientific) and Q Exactive HF mass spectrometer (Thermo Fisher Scientific). The analytical column RSLC PepMan C-18 (Thermo Fisher Scientific, 2 uM, 100 Å, 75 μm id × 50 cm) was used at 55 °C to analyze the samples with the mobile phase of buffer A (0.1% formic acid in MS grade water) and buffer B (90% acetonitrile in 0.1% formic acid) and injecting approximately 400 ng peptides. The liquid chromatography gradient was of 155 min from buffer A to buffer B at a flow rate of 300 nl/min having the following steps: 2 to 5% buffer B for 5 min, 5 to 25% buffer B for 110 min, 25 to 40% buffer B for 25 min, 40 to 80% buffer B for 5 min, and 80 to 95% buffer B for 5 min and an additional 5 min hold at 95% for Buffer B.

Further, the serum samples were processed using data independent acquisition (DIA) with the given parameters: for full-scan MS acquisition in the Orbitrap, the resolution was set to 120,000, having the scan range of 350 to 1650 m/z with the maximum injection time of 100 ms, and automatic gain control (AGC) target of 3 × 10^6^. The data acquisition was carried out using 17 DIA variable windows in the Orbitrap with resolution setting at 60,000, AGC target of 1 × 10^6^, and the maximum injection time in auto mode. The sample run order was randomized and with approximately every 6 samples, a quality control (QC) sample was run. The QC samples consisted of pooled serum samples processed in the same way as the above samples.

### 2.4. Protein Identification and Quantification

We used Spectronaut for all primary processing (v14, https://biognosys.com/software/spectronaut/). All 225 raw files were first converted to the HTRMS format with the HTRMS converter (centroid method). The converted files were then analyzed with the directDIA method using default settings. We exported intensity information at the fragment level for data preprocessing.

### 2.5. Proteomics Data Preprocessing

Following the extraction of values for fragment ion peak areas across all analysis files, we performed a series of quality control steps using in-house R scripts to remove analysis sequence and batch effects. These quality control steps were necessary to adjust for the impact of changes in the conditions of the mass spectrometry instrument while acquiring the data, both locally and globally along a randomized analysis sequence. We adapted this procedure from practices used in the processing of metabolomic mass spectrometry data [17].

Specifically, we first removed duplicate ions and ions with <20% presence across the samples, as well as equalized the median intensities of the samples. Then, we log_2_-transformed the intensity values of the ions, applied Gaussian kernel smoothing to remove temporal changes within each batch with a bandwidth of 5 samples (one standard deviation of the kernel) and equalized the median intensities across different batches again. Batches were defined based on experimental information on the run order, changes in the chromatographic columns, as well as the behavior of the quality control samples. The procedure has recently been implemented in the MRMkit software [18], and it removes systematic variation across samples by smoothing ion intensities of each analyte over the analysis time. Data smoothing can also be performed using other methods such as linear smoothers [19,20] or local regression approaches [21].

Next, we transformed the data back to linear scale and applied mapDIA [22] to select ions with consistent within-protein, inter-fragment correlation patterns across the samples to calculate the protein intensity. In mapDIA, any fragment ion with an average correlation value against all other fragment ions below 0.0 was removed before quantification. The removed ions can be considered as qualifiers (to identify the protein), but not as quantifiers. Finally, we normalized the protein intensities to the total intensity measured in each sample, performed quantile normalization, and log_2_-transformed the normalized intensities. These values were used for all statistical testing and functional analyses.

For visualization of the data in heatmaps, we further row centered the data by subtracting the row median intensity across samples from the intensity value of each protein. We performed principal component analysis on the row median centered data with the prcomp() function in R (https://www.R-project.org/, accessed on 4 November 2022). We found that the major (first) principal component was not associated with any of the demographic data, adjusted seroconversion or seroconversion, suggesting that its variability arose from other factors (Appendix A). For visualization purposes, we subtracted the first principal component and row Z score normalized the data.

### 2.6. Statistical Testing

For the analysis, we used a two-sample *t*-test [23] to compare the difference in protein levels between participants with top and bottom 30 raw or adjusted seroconversion values. We used the same test to compare the protein levels between participants with top and bottom 30 adjusted seroconversion values for the four individual strains (Appendix A), as well as to compare the protein levels between lean and obese participants (Appendix A).

We further performed Gene Set Enrichment Analysis (GSEA) with the web-based tool WebGestalt (http://www.webgestalt.org/, accessed on 4 November 2022), to identify enriched pathways using different annotations including GO, KEGG, and Reactome. As input for GSEA, we used for each protein the difference between the log transformed median abundance values across the samples from the 30 participants with the highest and lowest seroconversion. We kept pathways with a false discovery rate ≤ 0.05.

For the analysis, we first performed a two-way ANOVA involving two age groups and two adjusted seroconversion groups. Specifically, we defined young participants as ≤35 years (*n* = 20) and older participants as ≥60 years (*n* = 30). Within each group, top and bottom halves of the participants were defined as groups of high and low adjusted seroconversion, respectively. We extracted the *p*-values for the age-seroconversion interaction term, corrected for multiple hypothesis testing with the Benjamini-Hochberg procedure [24] and obtained adjusted *p*-values.

For easier visualization and interpretation of the results from the statistical testing, we converted adjusted *p*-values as follows: we first estimated the direction of the abundance change between two groups, e.g., young and old participants, comparing average protein levels. We then calculated a new value (1-Q) which received a positive and negative sign if the change in protein levels between two groups was positive or negative, respectively. We tested the GO, KEGG, and Reactome annotations and retained pathways with a false discovery rate ≤ 0.05 (Appendix A). Finally, we also performed a correlation-based analysis in which we used all samples to test for significant biases. Specifically, we correlated the abundance of each protein across all 138 samples with the unadjusted or adjusted seroconversion values, using Spearman’s correlation. The results were largely similar to those shown below (Appendix A).

## 3. Results

### 3.1. Quantification of ~300 Proteins in Influenza Vaccine Recipients

We selected serum samples from a large cohort study investigating the response to the split-inactivated influenza vaccination in the 2019–2020 season. Samples included healthy adult, Caucasian participants across the age range (Appendix A). Participant information had been removed except for information on participant age, body mass index (BMI), sex, race, existing comorbidities, prior vaccination status, month of vaccination, vaccine dose, as well as hemagglutination inhibition assay (HAI) titer levels against each of the four vaccine strains at day 0 and day 28 post vaccination. This information is provided in Appendix A.

We used the provided titer information to calculate seroconversion, i.e., the quantitative response to the vaccine as described elsewhere [16]. We then used the computational model we developed [9] to obtain the ‘adjusted seroconversion’ for each participant which provides a measure of seroconversion that removes the impact of age, BMI, sex, and the other factors mentioned above. Due to this correction, adjusted seroconversion can report vaccine effects independent of known factors such as age and prior vaccination status; it enables discovery of new factors. Appendix A shows the participant information ordered with respect to the adjusted seroconversion. It illustrated that unadjusted and adjusted seroconversion correlate, but adjusted seroconversion is largely independent of other factors. The unadjusted seroconversion is confounded by various factors such as age and baseline titer level, showing no or only minor association with the above factors (Appendix A).

We used mass spectrometry based proteomics to profile the protein levels in the serum collected on the day of vaccination (day 0). We analyzed samples from a total of 160 vaccine recipients of which 151 serum samples passed our quality control assessment. Of these, 138 samples had corresponding information on both unadjusted and adjusted seroconversion and were used for the analyses described below. We also measured protein levels in 65 quality control samples. Expression levels for the 151 samples from vaccine recipients are reported in Appendix A.

Figure 1 shows the protein levels across the 138 samples from vaccine recipients ordered from low to high unadjusted seroconversion, also displaying the additional participant information. The upper panel illustrates the impact of various factors on adjusted seroconversion. For example, it demonstrates the impact of prior vaccination on seroconversion: virtually all of the bottom half of seroconverters received the vaccine in the previous year. In comparison, participants with high seroconversion often had not been vaccinated in the year prior to the study.

The heatmap in Figure 1 (lower panel) illustrates the variability in protein levels measured across participants: there are no obvious patterns beyond small isolated clusters. Therefore, we confirmed the ability of the data to provide meaningful estimates of protein levels by investigating the association of protein levels with the demographic factors. We observed that the protein levels were most strongly associated with BMI. We found that leucine-rich alpha-2-glycoprotein (LRG) and complement factor H-related protein 4 (FHR4) are upregulated in obese participants compared to participants of normal weight (Appendix A). High levels of LRG and FHR4 are consistent with an upregulation of insulin resistence as is the case in the obese population [25]. This result supported our confidence in the quality of the data.

The upper panel illustrates the distribution of adjusted seroconversion, seroconversion and demographic information for all study participants. Serostatus is defined as in Wu et al. [9]. The lower panel shows the normalized protein levels for 273 proteins measured in participant sera. BMI, body mass index. Adj. seroconversion, adjusted seroconversion. Feb., February. Prevacc., Participants with prior vaccination. Sep., September. Vacc., vaccination.

### 3.2. High Seroconversion Associated with Elevated Cholesterol Metabolism and Actin Cytoskeleton Pathways

We compared the protein levels in participants with high and low unadjusted and adjusted seroconversion (Figure 2A,B). The results of the statistical testing, i.e., *p*-values and adjusted *p*-values, are provided in Appendix A. The overall signal (difference in protein levels) was larger for unadjusted seroconversion, likely due to the additional impact of confounding factors.

While none of individual proteins displayed statistically significant differences in protein levels after multiple hypothesis correction (Appendix A), gene set enrichment analysis (GSEA) identified significantly enriched pathways (false discovery rate <0.05): proteins from cholesterol metabolism had significantly higher levels in participants with high seroconversion (Figure 2A). Proteins in this pathway include apolipoproteins (APO-A1, A2, C2, E, A4 and C3), as well as cholesteryl ester transfer protein (CETP), lecithin-cholesterol acyltransferase (LCAT), and lipoprotein(a) LPA. The proteins are labeled in red in Figure 2A and their expression patterns are shown in Figure 2C. While again there is substantial variability in the protein levels, one can observe consistent differences in protein levels between participants with high and low seroconversion.

Similar analyses identified different significantly enriched pathways when analyzing adjusted seroconversion: proteins regulating the actin cytoskeleton were significantly upregulated in participants with high adjusted seroconversion (false discovery rate <0.05, Figure 2B). While not statistically significant at the individual protein level, the trend was clearly visible for the pathway members such as cofilin-1 (CFL1), profilin-1 (PFN1), and beta-actin (ACTB) (Figure 2D). The results in Figure 2 were confirmed using an alternative approach based on correlation analysis (Appendix A).

### 3.3. Association of Complement System, HDL Remodeling, Gene Transcription, and Hemostasis Pathways with Vaccine Response Changes with Age

We further tested whether there are more complex relationships between adjusted seroconversion and known demographic factors (Methods, Appendix A). Due to the association of several demographic factors with unadjusted seroconversion, such analysis was not meaningful using this measure. Again, we did not observe statistical significance at the level of individual proteins, but significant associations between age and adjusted seroconversion for several gene function pathways (false discovery rate < 0.05, Figure 3A). Since adjusted seroconversion has already been corrected for the linear impact of age, the observed signal is due to either a truly newly identified relationship between the vaccine response and recipient age, or due to the remaining link between adjusted seroconversion and age.

Specifically, we observed that proteins from the complement system have higher levels both in young participants with high seroconversion and in older participants with low seroconversion, and lower levels in the other groups (Figure 3B). We observed the opposite pattern for proteins from high density lipoprotein (HDL) remodeling, gene transcription, and hemostasis pathways. As above, the additional information gained from the functional association between genes (using GSEA) was able to extract this signal with statistical confidence (false discovery rate < 0.05).

### 3.4. Strain-Specific Differences in Vaccine-Response-Associated Protein Levels Changes

Finally, we examined the protein levels differences at a strain specific level associated with raw and adjusted seroconversion against each vaccine strain, e.g., A/H1N1, A/H3N2, B/Yamagata, and B/Victoria. The results from the statistical testing are provided in Appendix A. We observed that there was little overlap between proteins differentially abundant between low and high responders across the four strains, except for proteins that had higher levels in participants with high seroconversion for A/H1N1 and B/Victoria (Figure 4A). Proteins shared between these two sets included immunoglobulin kappa variable 4 (IGKV4), platelet factor 4 variant (PF4V1), pro-platelet basic protein (PPBP), angiotensinogen (AGT), lecithin-cholesterol acyltransferase (LCAT), and cholesteryl ester transfer protein (CETP) (Figure 4B,E).

Further, we also observed that proteins regulating the actin cytoskeleton such as CFL1 and PFN1 were among the top differentially abundant proteins in the A/H3N2 data, but they are not differentially abundant in the other three strains’ data (Appendix A). This result implies that the CFL1/PFN1 signal observed for composite adjusted seroconversion was largely driven by the A/H3N2 strain.

## 4. Discussion

We present one of the first large-scale analyses of the serum proteomic response to vaccination with the split-inactivated influenza vaccine. Our analysis has two components that enabled extraction of signals from highly variable data: (i) we used a quantitative measure of the vaccine response that was corrected for the impact of confounding factors such as age, BMI, or prior vaccination (adjusted seroconversion); (ii) we examined differential protein levels using additional information from gene pathway membership which increased signal sensitivity beyond what could be observed from testing individual proteins.

Using traditional response measures, such as unadjusted seroconversion, we found that low levels of proteins from cholesterol metabolism associated significantly with a low vaccine response (false discovery rate < 0.05), specifically several APOA and APOC proteins which function in reverse transport of high density lipoproteins in cardiovascular and hepatic disease [26]. Cholesterol metabolism is thought to influence innate and adaptive immune responses, possibly through its impact on membrane fluidity that affects the migration and communication of immune cells with each other [27]. However, the association between low levels of proteins from cholesterol metabolism and low seroconversion might also be due to confounding effects: cholesterol metabolism is linked to metabolic health of the individual, which is approximated here by BMI. BMI in turn, as well as age, associated negatively with the unadjusted seroconversion: obese and older participants had lower seroconversion than lean and young participants (Appendix A). The biases observed might therefore be explained by hypercholesterolemia or other age-related comorbidities or age itself as it can lower the efficiency of cholesterol metabolism [28].

Therefore, we complemented this analysis with the analysis of adjusted seroconversion, as this measure assesses the immune response while already correcting for other factors such as age and BMI, but also sex, race, vaccine dose, prior vaccination, month of vaccination, and baseline titer levels [9]. Using similar methods as for unadjusted seroconversion, we identified proteins regulating the actin cytoskeleton to have significantly higher levels in participants with high adjusted seroconversion than in participants with lower adjusted seroconversion. We showed that this signal might be primarily driven by the response to the A/H3N2 strain. We hypothesize that the dominant role of the A/H3N2 strain might arise from rapid divergence which causes the respective vaccine to differ more from season to season and therefore render a stronger response compared to those for other strains [29].

Three proteins were assigned to this pathway, namely beta-actin, cofilin 1, and profilin 1 (ACTB, CFN1, and PFL1). While these proteins are primarily known for their intracellular roles, several studies have described their marked abundance changes in serum. For example, cofilins 1 and 2 are serum markers of cancer progression as well as Alzheimer’s disease [29,30]. Dysregulation of profilin 1 level in the serum is associated with diabetes and implicated in cancer and cardiovascular diseases [31]. The cause and consequences of this association of beta-actin, cofilin 1, and profilin 1 with adjusted seroconversion are unclear, but there are known cues to infer the relationship. For example, the actin cytoskeleton has a key role in the immune responses [32], and there is an emerging role of functional mutations or expression dysregulation of the actin cytoskeleton in primary immunodeficiency disorders [33,34]. A possible mechanism lies in the key role of actin cytoskeleton dynamics in the formation of the immunological synapse during T-cell activation [32].

An even more speculative interpretation relates to emerging links between viral infection and platelet counts [35,36]. Platelet activation involves, amongst many other pathways, profilin 1, cofilin, and the actin cytoskeleton, and viral infection can temporarily lower platelet counts [36]. Low PFN1, CFN1, and ACTB levels in individuals with low adjusted seroconversion might indicate a recent viral infection (prior to receiving the vaccine) which weakened the immune response. Future work will have to test this hypothesis.

Finally, we observed an interaction between vaccine recipient age and adjusted seroconversion, resulting in a complex pattern (Figure 3). We found that proteins from the complement system have higher levels both in young participants with high seroconversion and in older participants with low seroconversion. This finding is consistent with a study of the transcriptome in PBMCs which identified the same pattern for genes of the inflammatory response [14]. As the inflammatory response intersects with the complement system described here (Figure 3), the study confirms our findings. Another study found that proteins of the complement system are among the top proteins with a significant glycan Le^a^ difference between low and high influenza vaccine responders, implying that indeed the complement system affects the vaccine response [15].

In all, our analysis identified regulation of the actin cytoskeleton and the complement system as pathways associated with the response to influenza vaccination—independent of trivial factors such as age and body mass index. While further studies will be needed, the findings suggest novel routes that might impact the vaccine response.

## Figures and Tables

**Figure 1 viruses-14-02479-f001:**
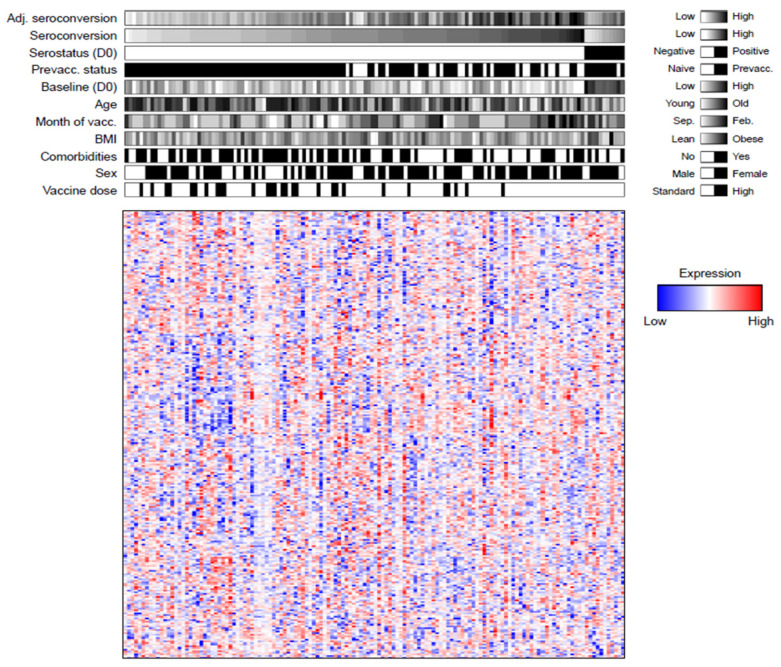
Protein levels of 273 proteins across recipients of the split-inactivated influenza vaccine.

**Figure 2 viruses-14-02479-f002:**
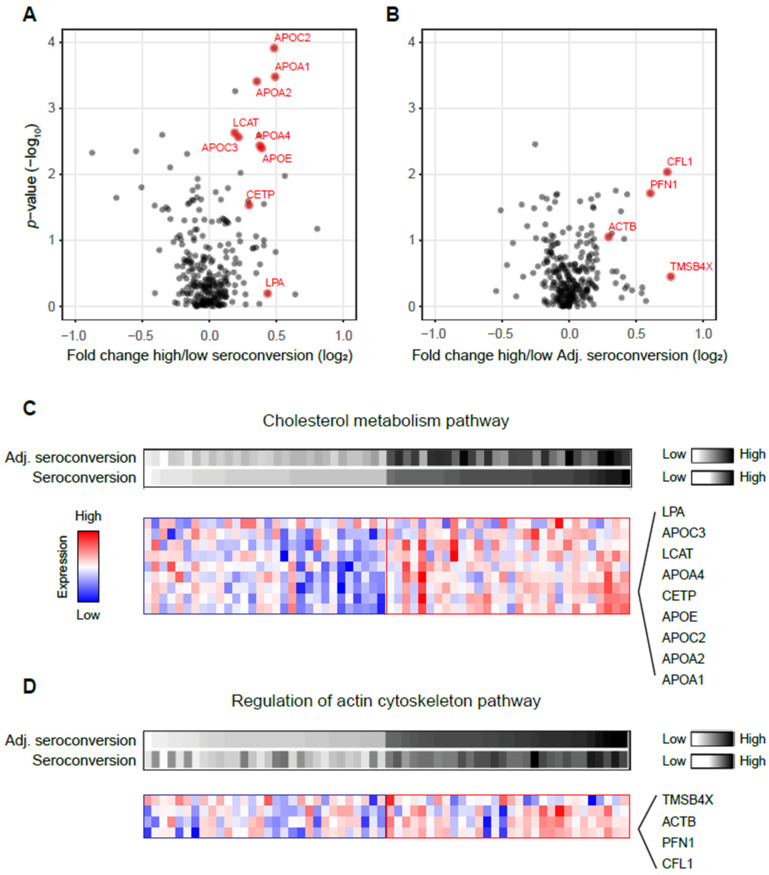
Cholesterol metabolism and regulation of actin cytoskeleton pathways are upregulated in participants with high seroconversion before and after adjusting for confounding factors, respectively. (**A**) Volcano plot showing fold change difference in protein levels and corresponding *p*-value between participants with the highest and lowest seroconversion. (**B**) Volcano plot showing fold change difference in protein levels and corresponding *p*-value between participants with the highest and lowest adjusted seroconversion. In both panels, red marks leading proteins from significantly enriched pathways (see Methods). (**C**) Proteins of cholesterol metabolism have higher levels in participants with high seroconversion in response to the influenza vaccine than in participants with low unadjusted seroconversion. (**D**) Genes regulating the actin cytoskeleton have higher levels in participants with high adjusted seroconversion in response to the influenza vaccine than in participants with low seroconversion. Adj. seroconversion, adjusted seroconversion.

**Figure 3 viruses-14-02479-f003:**
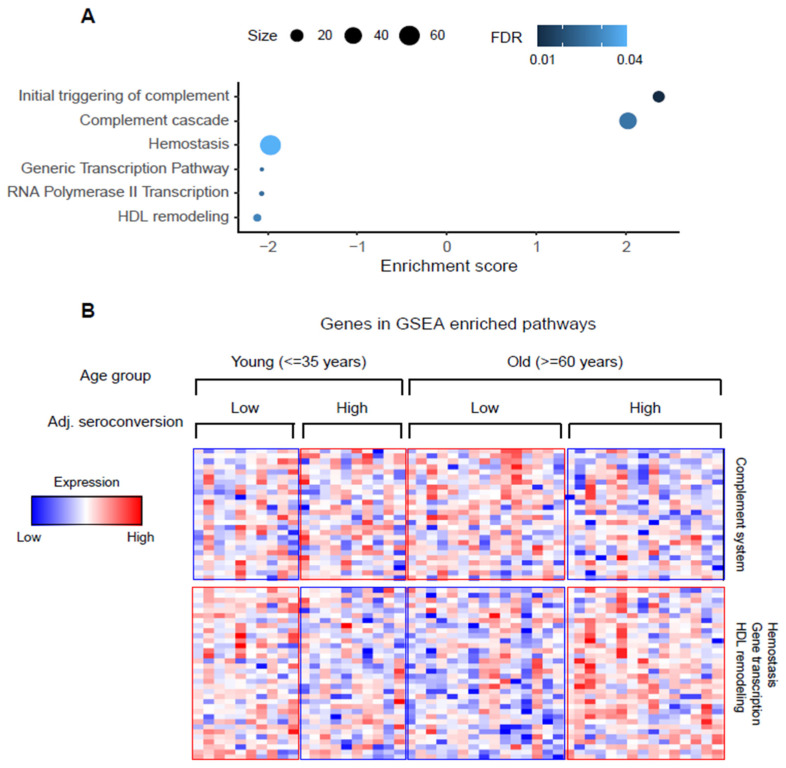
Genes of the complement system, HDL remodeling, transcription, and hemostasis have complex relationships with adjusted seroconversion and age of the vaccine recipient. (**A**) Gene functional pathways that are significantly enriched in the age & adjusted seroconversion two-way ANOVA (see Methods). (**B**) Levels of the proteins from the significantly enriched pathways. Adj. seroconversion, adjusted seroconversion.

**Figure 4 viruses-14-02479-f004:**
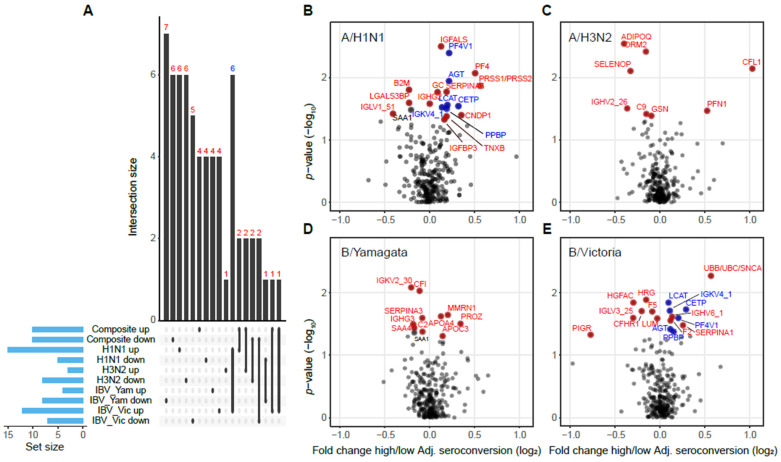
Strain-specific signatures of adjusted seroconversion. (**A**) Overlap between up- and down-regulated genes in participants with the highest and lowest composite and strain-specific adjusted seroconversion. Genes up-regulated in participants with high adjusted seroconversion against A/H1N1 and against B/Victoria have the largest overlap and are labeled in blue. (**B**–**E**) The volcano plots show the strain-specific fold changes in protein levels between participants with high and low adjusted seroconversion. Genes labeled in red and blue colors correspond to the sets shown in panel (**A**). Adj. seroconversion, adjusted seroconversion.

## Data Availability

We deposited the 225 raw files of the serum proteome containing serum samples from 160 influenza vaccine recipients and 65 quality control samples to PRIDE, with the accession number PXD036718. Spectronaut output, batch information, preprocessed and normalized data files, as well as the mapDIA parameter setting file and R scripts were deposited to Github: https://github.com/sw5019/Fluvacc-serum-proteomics-project, accessed on 4 November 2022.

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
