# Peer review of "Proteomic Signatures of the Serological Response to Influenza Vaccination in a Large Human Cohort Study"

_viruses, 2022, doi:10.3390/v14112479_

Round 1
Reviewer 1 Report
In this work, authors conduct integrative modeling to extract associations between protein-level signatures and serological response to influenza vaccine. This problem poses great interest both to the scientific community and public health authorities, as obtained insights can help to make new hypotheses on the molecular mechanism of action of the existing vaccines and help guide the design of the new solutions. Statistical methods used for the analysis are robust; the authors make necessary adjustments for the confounding factors. Data is deposited in specialized data repository PRIDE, and code is made publicly available on the GitHub.
The authors produced high-quality work. However, I have some comments that can help to better communicate their findings:
- The authors conducted a large-scale proteomics study on flu vaccine recipients that quantified 273 protein levels from 225 serological samples, both individual and quality control 160 samples from 138 individuals. This data is deposited in the PRIDE database (https://www.ebi.ac.uk/pride/archive/projects/PXD036718/private). Data description in PRIDE refers to Allgoewer K, 2021 as a data processing protocol. Most of the information on wet lab protocols is covered in sections 4.1 - 4.4. However, the composition of final dataset in Section 2.1 is not clear. I assume the matched samples take place in the analysis because Abstract and PRIDE mention 160 serological samples and Section 2.1 mentions 138 individuals; correct me if I am wrong. I think it would be valuable to give readers more context on the entire collection of 225 samples and clarify the role of matched samples in the analysis. Currently, readers have to combine information from the Abstract, Section 2.1, and Section 4 or data description from PRIDE to get those numbers, so a more systemized description of the dataset would be beneficial.
- The authors conducted GSEA analysis on the multiple pathways and adjusted statistical significance for FDR < 0.05. I have only two minor suggestions. First is to clarify whether the Benjamin-Hochberg FDR method was used for every multiple testing correction procedure. The second is to include GSEA outputs in the supplementary information, either in the form of the master table with statistics for all groups, or as a collection of images for individual pathways. I am not familiar with http://www.webgestalt.org/, so my suggestions are based on the GSEA implementation from Broad Institute. This information would be valuable for follow-up studies.
- Section 4.5 describes data preprocessing where the authors guide readers through the custom normalization process. I think it would be valuable for the audience to compare this procedure with the standard manufacturer’s protocol, highlight key differences, and elaborate on which problems the custom protocol solves. I hope it can help the scientific community to identify pitfalls in proteomics data processing and improve the quality of subsequent studies.
Author Response
Reviewer 1’s comments:
In this work, authors conduct integrative modeling to extract associations between protein-level signatures and serological response to influenza vaccine. This problem poses great interest both to the scientific community and public health authorities, as obtained insights can help to make new hypotheses on the molecular mechanism of action of the existing vaccines and help guide the design of the new solutions. Statistical methods used for the analysis are robust; the authors make necessary adjustments for the confounding factors. Data is deposited in specialized data repository PRIDE, and code is made publicly available on the GitHub.
The authors produced high-quality work. However, I have some comments that can help to better communicate their findings:
- The authors conducted a large-scale proteomics study on flu vaccine recipients that quantified 273 protein levels from 225 serological samples, both individual and quality control 160 samples from 138 individuals. This data is deposited in the PRIDE database (https://www.ebi.ac.uk/pride/archive/projects/PXD036718/private). Data description in PRIDE refers to Allgoewer K, 2021 as a data processing protocol. Most of the information on wet lab protocols is covered in sections 4.1 - 4.4. However, the composition of final dataset in Section 2.1 is not clear. I assume the matched samples take place in the analysis because Abstract and PRIDE mention 160 serological samples and Section 2.1 mentions 138 individuals; correct me if I am wrong. I think it would be valuable to give readers more context on the entire collection of 225 samples and clarify the role of matched samples in the analysis. Currently, readers have to combine information from the Abstract, Section 2.1, and Section 4 or data description from PRIDE to get those numbers, so a more systemized description of the dataset would be beneficial.
Reply: We edited section 2.1 on page 3, and Data Availability Statement on page 12 to clarify these issues: we had analyzed 160 samples from flu vaccine recipients, as well as 65 quality control samples. Of the 160 samples, 9 were removed based on poor data quality. Of the remaining 151 samples, 138 had information on both unadjusted and adjusted seroconversion available. These 138 samples were used for the analyses.
- The authors conducted GSEA analysis on the multiple pathways and adjusted statistical significance for FDR < 0.05. I have only two minor suggestions. First is to clarify whether the Benjamin-Hochberg FDR method was used for every multiple testing correction procedure. The second is to include GSEA outputs in the supplementary information, either in the form of the master table with statistics for all groups, or as a collection of images for individual pathways. I am not familiar with http://www.webgestalt.org/, so my suggestions are based on the GSEA implementation from Broad Institute. This information would be valuable for follow-up studies.
Reply: We included the GSEA output, i.e. all pathways reported with a false discovery rate < 0.05, in Supplementary Data File S1. We noted this in the text.
- Section 4.5 describes data preprocessing where the authors guide readers through the custom normalization process. I think it would be valuable for the audience to compare this procedure with the standard manufacturer’s protocol, highlight key differences, and elaborate on which problems the custom protocol solves. I hope it can help the scientific community to identify pitfalls in proteomics data processing and improve the quality of subsequent studies.
Reply: We expanded Section 4.5 on page 11 to include details on the normalization process, the rationale behind the different steps, as well as alternative methods available.
Reviewer 2 Report
- This manuscript examines levels of 273 proteins in 160 influenza vaccine recipients using seroconversion adjusted for demographic factors and found positive associations between levels of proteins of the complement system in younger participants and negative associations in the older population. Opposite trends were observed for proteins of high density lipoprotein remodeling, transcription, and hemostasis. This is an exploratory analysis, with no specific hypothesis stated.
- Comments:
- The authors have provided background support for their study and in-depth analysis of their findings. The results are presented in detail, however limited detail is provided on the study population beyond healthy adult Caucasian participants. While demographics are largely controlled for in the adjusted seroconversion value and represented in gray scale across recipients, it would be helpful to know more details on the population which could be represented in a demographic table. (This may be already be included in Supplementary Data File S1, but I cannot access it.) For example, age is represented as a continuum from young to old, BMI lean to obese, and co-morbidities yes/no. Mean values with confidence intervals and general definitions of what co-morbidities a population defined as 'healthy adults' might have. Since the co-morbidites relate to the seroconversion value, it would be interesting to know if they were selected based on effects on immune response, influenza risk, or both.
- Month of vaccination (September to February) ranges well into the typical influenza season. In comparison, most vaccinations are received in September through November and influenza cases most commonly peak in February. For those vaccinated later in the season, it is possible that the measured response could the the result of influenza infection. Explanation as to why vaccines were administered so late in the season and the impact of potential influenza infection on seroconversion and protein response is warranted.
- Supplementary data is referred to as 'Supplementary Data File S1' but I do not have access to this file to confirm the material that is being referred to.
- Minor point: Consider 'influenza' instead of 'flu' in title and throughout abstract and text for consistency.
Author Response
Reviewer 2’s comments:
This manuscript examines levels of 273 proteins in 160 influenza vaccine recipients using seroconversion adjusted for demographic factors and found positive associations between levels of proteins of the complement system in younger participants and negative associations in the older population. Opposite trends were observed for proteins of high density lipoprotein remodeling, transcription, and hemostasis. This is an exploratory analysis, with no specific hypothesis stated.
Comments:
- The authors have provided background support for their study and in-depth analysis of their findings. The results are presented in detail, however limited detail is provided on the study population beyond healthy adult Caucasian participants. While demographics are largely controlled for in the adjusted seroconversion value and represented in gray scale across recipients, it would be helpful to know more details on the population which could be represented in a demographic table. (This may be already be included in Supplementary Data File S1, but I cannot access it.) For example, age is represented as a continuum from young to old, BMI lean to obese, and co-morbidities yes/no. Mean values with confidence intervals and general definitions of what co-morbidities a population defined as 'healthy adults' might have. Since the co-morbidites relate to the seroconversion value, it would be interesting to know if they were selected based on effects on immune response, influenza risk, or both.
Reply: We included a table summarizing the demographics (Supplementary Data File S1) and refer to it in the text.
- Month of vaccination (September to February) ranges well into the typical influenza season. In comparison, most vaccinations are received in September through November and influenza cases most commonly peak in February. For those vaccinated later in the season, it is possible that the measured response could the the result of influenza infection. Explanation as to why vaccines were administered so late in the season and the impact of potential influenza infection on seroconversion and protein response is warranted.
Reply: We expanded the descriptions in Sections 4.1. and 4.2. on p. 10 to clarify the following points. Samples had been obtained from a pre-designed study in which volunteers provided blood samples upon receiving the influenza vaccine. The study design did not limit the time window (month of vaccination) during which the vaccine was administered. As the reviewer noted, ‘Month of vaccination’ did indeed positively affect the response: vaccines administered in later months triggered a stronger response. This was discussed extensively in our previous study which we now cite (Wu, Mol Systems Biology, 2022 - https://doi.org/10.15252/msb.202110724). We clarified in the manuscript that the calculation of ‘adjusted seroconversion’ aimed at removing exactly this and other confounding factors. Therefore, the impact of Month of vaccination (and other demographic variables) was largely removed when examining the adjusted seroconversion.
- Supplementary data is referred to as 'Supplementary Data File S1' but I do not have access to this file to confirm the material that is being referred to.
Reply: We have now included Supplementary Data File S1. We apologize for this oversight during the initial submission.
- Minor point: Consider 'influenza' instead of 'flu' in title and throughout abstract and text for consistency.
Reply: We edited the entire manuscript accordingly.
Round 2
Reviewer 2 Report
Thank you for addressing the concerns. I feel like your points are much clearer with the revision. Congratulations on a nice study.